# Goal-Framing and Temporal-Framing: Effects on the Acceptance of Childhood Simple Obesity Prevention Messages among Preschool Children’s Caregivers in China

**DOI:** 10.3390/ijerph17030770

**Published:** 2020-01-26

**Authors:** Qingmao Rao, Li Bai, Yalan LV, Abu Saleh Abdullah, Ian Brooks, Yunjie Xie, Yong Zhao, Xiaorong Hou

**Affiliations:** 1College of Medical Informatics, Chongqing Medical University, Chongqing 400016, China; rqm@stu.cqmu.edu.cn (Q.R.); 2017111298@stu.cqmu.edu.cn (L.B.); yalanlv@126.com (Y.L.); 2Global Health Research Center, Duke Kunshan University, Kunshan 215347, China; abu.abdullah@dukekunshan.edu.cn; 3Duke Global Health Institute, Duke University, Durham, NC 27710, USA; 4Boston University School of Medicine, Boston Medical Center, Boston, MA 02118, USA; 5Center for Health Informatics, University of Illinois, Champaign, IL 61820, USA; ianb@illinois.edu; 6School of Information Sciences, University of Illinois, Champaign, IL 61820, USA; 7College of Preschool Education, Chongqing University of Education, Chongqing 400067, China; xieyj@cque.edu.cn; 8School of Public Health and Management, Chongqing Medical University, Chongqing 400016, China

**Keywords:** goal-framing effects, temporal-framing effects, acceptance, health message, caregivers

## Abstract

A range of intervention models are available for childhood obesity prevention; however, few studies have examined the effectiveness of intervention messages. This study developed childhood simple obesity prevention messages on the basis of goal-framing and temporal-framing effects to improve message acceptance among the caregivers of preschool children and explored associated factors. A cross-sectional study was conducted among 592 caregivers of preschool children in urban kindergartens in China during March to April 2019. The framing messages were developed based on prospect theory and construal level theory. The majority (48.4%) of caregivers found the gain-framed, present-oriented message most salient for acceptance. We found that gender, education background, theme, and the use of negative words have impacts on goal-framing effects; and previous participation in a health related intervention, career category, and the theme have impacts on temporal-framing effects (*p* < 0.001). Goal-framing effects and temporal-framing effects can influence each other (*p* < 0.001). The findings suggest that the gain-framed, present-oriented message could be considered a strategy to improve the acceptance of information by caregivers. When framing a message, subtle differences like using negative words might affect the exertion of framing effects.

## 1. Introduction

Simple obesity is caused by excessive accumulation of fat as the body absorbs more calories than it consumes [1]. Childhood simple obesity was once considered a problem only in high-income countries, but is now dramatically on the rise in low and middle-income countries, particularly in urban settings [2]. According to the data provided by the International Obesity Task Force and the World Health Organization disease burden report, the rate of global childhood simple obesity is currently 2%–3%, and the overall trend is on the rise in all regions [3,4]. Geserick et al. [5] found that the preschool years are a critical period for childhood simple obesity, and the rapid increase in body mass index (BMI) among children aged 2 to 6 years old was highly correlated with obesity in adolescence and even adulthood, with a relative risk of 1.43. Simple obesity among preschool children can be regarded as a major factor for several chronic diseases, such as hypertension, dyslipidemia, and diabetes [6,7]. Therefore, it is necessary to prevent simple obesity among children and intervene at the preschool stage (3–6 years of age [8]).

Childhood simple obesity is also a typical lifestyle disease [9]. Unhealthy diets such as excessive amounts of sweets and unhealthy behavior, e.g., picking food to eat, as well as a sedentary lifestyle, may lead to the development of preschool childhood obesity [10,11]. Especially under the influence of traditional Chinese family culture, the caregivers often reward their children with snacks, such as candy, and encourage them to eat more food; this phenomenon is common in the skip-generation raising families [12]. Parents, grandparents, and other caregivers prepare the food of preschool children. Many Chinese scholars have conducted several studies in various regions of China and found that increasing the level of health knowledge of the caregivers can encourage children to have a positive attitude toward health and to maintain a healthy lifestyle; thus, the prevention and control of preschool children’s simple obesity can be improved [13,14,15]. This scenario underscores the need for family intervention in preventing childhood simple obesity.

Caregivers’ acceptance of health promotion messages is limited [16,17]. The difference between a health message and the general message is that the former is a normative, objective, and rigorously written medical material. Caregivers with different cognitive abilities and from different professional education levels may have considerable differences in the degree of and the time they need for acceptance [18]. At present, health message acceptance can be improved by changing the presentation form of materials, such as by using appropriate pictures, simple grammar and vocabulary, and concise presentation [19,20,21]. The current study aimed to explore the acceptance effects of four commonly used health communication message formats (i.e., gain-framed, loss-framed, present-oriented, and future-oriented).

The “framing effect” was first proposed by Tversky and Kahneman in 1981 [22]. Several studies have shown that message framing is a reliable way to improve the messaging utility, and adopting an appropriate frame for a message on a particular theme can enhance the acceptance of this message by the target population [23,24,25]. Myers [26] thinks that the health framing effect plays an important role in promoting healthy behavior. 

In framing effects, the goal-framing and temporal-framing effects are especially emphasized. Goal-framing effects based on prospect theory suggest that factually equivalent messages have different levels of persuasiveness depending on how these messages are framed. A gain-framed message focuses on the positive consequences of performing an action, whereas a loss-framed message focuses on the negative consequences of inaction [27]. Rothman [28] and most scholars argued that a gain-framed message is acceptable for disease prevention, whereas a loss-framed message is compelling for disease detection. However, some scholars had drawn opposite conclusions [29]. Temporal-framing effects are based on construal level theory, which proposes that people use abstract mental representations to perceive events in the distant future but not in the near future [30]. A message can be classified into present-oriented and future-oriented frames depending on the time perspective. The present-oriented framing effect was stronger than the future-oriented one, as determined by many researchers [31,32,33]. 

Therefore, it is necessary to prevent and intervene in simple obesity and intervention among children during the preschool stage on the basis of goal and temporal-framing effects. This research explored whether the acceptance of prevention messages regarding simple obesity in preschool-aged children can be increased by gain-framed messages and present-oriented messages among caregivers, and demonstrated the roles of demographic characteristics, themes of healthy behavior, personal involvement, and use of negative words within the message acceptance of moderating framing effects.

## 2. Materials and Methods 

### 2.1. Study Design

A cross-sectional study about the caregivers’ acceptance of the different framings of prevention messages regarding simple obesity in preschool-aged children was conducted in urban kindergartens in China from March to April 2019. We adopted convenience sampling to select eight urban kindergartens, which were located in Chongqing, China. A self-administered paper questionnaire and a self-filling online questionnaire were used. We also explored whether gender, education background, theme, the use of negative words, previous participation in a health related intervention, and career category might affect the recipients’ acceptance of framing messages.

### 2.2. Participant Selection

The following were the inclusion criteria. The participants must be actual caregivers of preschool children, who are defined as the people responsible for the diets and exercise levels of preschool children who assume the duty of guardianship of those preschool children. 

We recruited 312 participants through an online survey. Eight kindergarten teachers supported data collection. These teachers were responsible for posting an online link to the questionnaire, a message containing the survey purpose and process, and instructions for filling in the questionnaire in the WeChat group of preschool children’s caregivers. WeChat is a free application that provides instant messaging services for smart terminals. Caregivers who were interested in this survey could click on the link and fill in the questionnaire.

We recruited 280 participants through an offline (face-to-face) survey. We sent three trained investigators to the kindergartens for data collection.

### 2.3. Questionnaire

We referred to the review articles [24,34] and research papers [35,36,37] of framing effects to determine the questionnaire content of demographic characteristics. The contents of the prevention messages regarding simple obesity in preschool children were derived from clinical research [38,39], epidemiological studies [40], and clinical practice guidelines [41,42,43]. We processed these prevention messages into health-framed messages according to the prospect theory and construal level theory (Appendix A). The content of the questionnaire was finalized after several discussions by an expert group. In March 2019, the pre-survey investigated 31 caregivers to test the reliability of the questionnaire. The Cronbach’s alpha of the questionnaire (demographic characteristics and framing messages) was found to be 0.817. The Cronbach’s alpha for the framing messages materials was 0.893. 

Questions included caregiver’s relationship with the children, gender, age, ethnicity, educational background, type of work, whether or not health interventions for childhood obesity had been before, the monthly cost on the child’s diet, and multiple questions on message framing. Participants observed four different types of framing message on the same theme at the same time. They were asked to read the messages carefully and chose the most acceptable one (Appendix B).

The design process of the framing message was as follows. We first identified four prevention themes for obesity among preschool children: dietary habits (Dh), dietary behavior (Db), physical activities (Pa), and sleep factors (Sf). Then, we designed four sets of framing messages in four themes: a gain-framed, present-oriented message (GP message); a gain-framed, future-oriented message (GF message); a loss-framed, present-oriented message (LP message); and a loss-framed, future-oriented message (LF message) (Figure 1).

### 2.4. Data Collection 

Demographic questions were answered by the participants. Meanwhile, we allowed the participants to choose the framed message that most agreeable among the four. The answer to the online questionnaire was recorded automatically on the client-side. The answers to the offline questionnaire were provided by participants and were imported into an Excel file by the interviewers.

### 2.5. Ethical Aspects

The study was approved by the ethics committee of Chongqing Medical University (approval number: 2018011). The participants provided informed consent for inclusion before joining this study.

### 2.6. Statistical Analysis

Data were processed by Excel software before entry into the database. Data analyses were performed using SPSS 20.0 software (IBM Corporation, Armonk, NY, US). For the demographic characteristics of caregivers, frequencies and percentages were calculated for categorical variables and means and standard deviations for continuous variables. Percentages were used to describe caregivers’ choices in framing messages. Pearson’s chi-square test and the Bonferroni method were used to analyze the differences in caregivers’ choice tendencies of framing messages among the four different themes. Statistical analyses were conducted using a two-sided test. Binary logistic regression analysis was implemented to analyze the factors associated with framing effects. A *p*-value not more than 0.05 was considered statistically significant.

### 2.7. Quality Control 

The offline research team members, including three students (two postgraduate and one undergraduate), received standardized investigation training. Investigators were to understand the purpose and methodology of the study in detail and have extensive experience in dealing with potentially sensitive issues. In online research, the quality of the questionnaire answers can be guaranteed through screening of answer results and answering time.

## 3. Results

A total of 592 subjects participated in the survey. After excluding the invalid questionnaires (87), 505 sets of data from caregivers were available for analyses with a valid participation rate of 85.3%.

### 3.1. Demographic Characteristics of the Caregivers

The demographic characteristics of the samples are presented in Table 1. The majority of caregivers were the mothers of children. The caregivers’ mean age was 39.1 years old (11.07 SD). The Han ethnicity constituted 96.0% of the sample. This survey was basically carried out in urban kindergartens, so 258 (51.1%) of the caregivers had a university or three -year college education. The job categories of the caregivers in decreasing order were as follows: 166 administrative workers, soldiers, teachers, medical staff members, and scientists (32.9%); 124 workers in commerce (24.6%); 91 workers (18.0%); 69 unemployed (13.7%); and 18 farmers (3.6%). Among the caregivers, 260 (51.5%) allot between 500 and 1000 CNY for their children’s monthly dietary expenses. More than half (75.2%) of the caregivers did not previously participate in any health interventions for preschool childhood obesity voluntarily; an example would be receiving relevant health education. We compared the demographic characteristics of online and offline groups and found that the career category of the online group was dominated by administrative organs, soldiers, teachers, medical staff, scientists, and commerce workers, while the career category of the offline group was dominated by workers and commerce workers. In the online group, the number of caregivers who were grandparents was a little lower than that of the offline group, and the number of caregivers who were female, of high educational background, and paid high expenses for their children’s diets, was higher than that of the offline group.

### 3.2. Caregivers’ Choices of Framing Messages

The caregivers’ acceptance of framing messages is shown in Table 2. The majority of the caregivers found that the GP message was highly salient for acceptance in terms of dietary habits, physical activities, and sleep factors. However, more than half (50.1%) of the caregivers found the LP message most acceptable in terms of dietary behavior. 

We analyzed the two groups of framing messages separately to better analyze the effects of the goal-framing and temporal-framing effects. As shown in Table 3, in the goal frame message, a significant difference was found in the dietary behavior theme (“b) and other themes (“a”) (*p* < 0.05). In the temporally-framed message, a significant difference was observed in the dietary habits (dietary behaviors) theme (“a”) and the physical activity (sleep factors) theme (“b”) (*p* < 0.05). 

### 3.3. Binary Logistic Regression Analyses for the Factors Influencing the Framing Effects

To further investigate the factors that affect the acceptance of framing messages, the study performed binary logistic regression analysis. According to the results of Table 3 and previous studies, we chose twelve parameters as independent variables. The two logistic models were statistically significant. (χ^2^ = 558.353, *p* < 0.001; χ^2^ = 224.482, *p* < 0.001).

As shown in Table 4, in the goal framing model, gender, education background, theme, the use of negative words, and temporal framing were statistically significant (*p* < 0.05). Compared with the Dh theme, caregivers were more willing to accept the gain-framed message in Sf theme. The combination of the message and the present-oriented frame made caregivers more willing to accept the gain-framed message. Being female, having a lower educational background, the Db theme, and the existence of negative words in a message made caregivers more willing to accept the loss-framed message respectively.

As shown in Table 5, in the temporal framing model, whether one participated in a health related intervention before, career category, theme, and goal framing were statistically significant (*p* < 0.05). Caregivers with the experience of a health related intervention were more willing to accept the future-oriented message. Compared with the unemployed caregivers, peasant caregivers were more willing to accept a present-oriented message. Compared with the Dh theme, caregivers were more willing to accept a future-oriented message in the Pa, Sf theme. The combination of the message and gain-framed frame made caregivers more willing to accept a present-oriented message.

## 4. Discussion

Few previous studies have explored message framing for caregivers of preschool children to address simple obesity. In the study, we found that the GP message showed salience for acceptance in terms of dietary habits (385, 76.2%), physical activities, (304, 60.2%), and sleep factors (276, 54.7%). In terms of dietary behavior, 253 (50.1%) caregivers found the LP message acceptable. We synthesized the four theme messages and found that the gain-framed message was more acceptable than the loss-framed message (gain versus loss = 1503:517); the present-oriented message was more acceptable than the future-oriented message (present versus future = 1530:490). Then, we discussed some factors that would affect the framing effects through statistical analysis.

Firstly, using negative words in the message may affect the goal-framing effects. The result in Table 2 showed that GP message was most acceptable by caregivers except in Db. The results of further analysis are shown in Table 3; in terms of Dh, Pa, and Sf, the gain-framed message was more acceptable, whereas for Db, the loss-framed message was more acceptable (*p* < 0.05). As shown in Table 4, the existence of negative words and Db theme made caregivers more willing to accept the loss-framed message than the gain-framed message (*p* < 0.05). In order to explore the reason, we compared the characteristics of sentences under each theme. The framing messages were commonly designed as sentences, consisting of presuppositions of conditional propositions (antecedent framing) and consequent propositions (consequent framing) [25,44]. The affirmative sentences were used as conditional sentences, and the profit phenomenon was described as a resulting sentence, which formed a gain-framed message. By contrast, the negative sentences were used as conditional sentences, and the loss phenomenon was described as consequential sentences, thereby forming a loss-framed message. In Dh, Pa, and Sf theme, we followed these rules. But watching TV is a risk factor for obesity in dietary behavior (Db); we used negative sentences as conditional sentences to form a gain-framed message (matching a profitable result) and affirmative sentences as conditional sentences to form a loss-framed message (matching a loss result). The comparison of text materials is shown in Appendix A. The results showed that the different choices caregivers made in Db theme are probably related to the collocation conflict between negative words and frames. The inclusion of negative words in sentences slows down message processing and predisposes receivers to errors [45,46]. Meanwhile, using negative and affirmative sentences may lead to different psychological processes [47]. In summary, we suggest that further study on the relationship between negative words use and framing effects could help us better understand framing effects.

Then, with the increase of temporal framing in a message, the effect of temporal framing might become weaker. Traditional psychological studies have long noted the human tendency to discount temporally distant consequences [48]. Temporal-discounting research also shows that present-oriented consequences have more psychological value than future-oriented consequences [49]. Our overall research results were consistent with this finding. However, the influence of the temporal-framing effects on caregivers was different in four different thematic messages (Table 3 and Table 4). It is worth noting that when filling in the questionnaire, caregivers were required to read framing messages in existing order and choose the most acceptable message in one theme according to the order in which they appeared. The order of the theme was Dh (number 1), Db (number 2), Pa (number 3), and Sf (number 4). The results show that there was a trend of increasing the proportion of caregivers who thought the future-oriented message was more acceptable, with the increase of the serial number corresponding to the theme. With the increase of the temporal framing of a message, the proportion of present-oriented and future-oriented message selection approaches 1. Construal level theory is used to explain the temporal framing effects [30]. People tend to underestimate the value and importance of future events and outcomes because a future-oriented message is related to abstract psychological representations [50,51]. Therefore, a future-oriented message creates a low perceived risk. But our results suggested that increasing the number of framing messages may enable individuals to examine and assess the differences between future and present-oriented messages, and this tendency to underestimate the future is ultimately reduced. We thought this was a good way to study temporal framing effects.

At last, many factors might affect the frame effects. First, gender is an important factor influencing the goal framing effects. Huang et al. [37] found that women were more likely to be affected by the loss-framed messages when it came to life and death. Our results also show that the female group is more reluctant to accept the gain-framed message, which may be due to the pressure and responsibility of raising children [52]. Second, the influence of knowledge level on framing effects is not conclusive. Our result was similar to the result of Haider et al [35]. That is to say, people with low education were more likely to choose the gain-framed message under the influence of goal framing effects. Third, is whether participation in a relevant health intervention is the embodiment of personal involvement [53]. Many kinds of literature have confirmed that participants with higher personal involvement are more likely to be influenced by the goal-framing message [25]. However, the interaction between personal involvement and temporal framing shows a lack of detailed research. Time preference changed dynamically with the change of human sensitivity to duration [54]. The result of our study may suggest that getting the relevant knowledge in advance might reduce the sensitivity to time, thereby making caregivers more willing to accept the future-oriented message. It would be a good idea to research temporal framing effects. Fourth, farmers tend to maintain more traditional health attitudes and eating habits. Our result suggested that the present-oriented message might be applied in health education in rural areas. These results showed that when publicizing the message of children’s obesity in a specific population with caregivers, the more suitable framing message can be selected according to the characteristics of the caregivers.

This study has several limitations. First, we adopted a convenient sampling method in the experimental design to ensure that the subjects included in the study were willing to read our information materials carefully. However, we increased the contingency and reduced the typicality of the samples. Second, because our survey was conducted in urban public kindergartens, the number of individuals with different demographic characteristics varied. In particular, there was a lack of rural participants. At the same time, there were not enough grandparents, males, and individuals with low educational backgrounds. Therefore, the sample of these groups was less representative of choosing the framing messages. It is worth noting that there were differences in demographic characteristics between the online and offline groups. Although the survey method had no statistical impact on the selection of framing messages, it still suggested that the online questionnaire has limitations on the selection of the population. Third, in the framing messages design, the use of negative expression under the dietary behavior theme was different from that under other themes, and only this group used a negative expression in the gain-framed message. We did not have a strict control experiment on the use of negative words. Fourth, due to the lacking literature regarding farmers and framing effects, our discussion on farmers has a lack of scientific basis, which reduces its credibility. Finally, the study relied on self-reporting, which can introduce bias because of dishonesty, over-reporting, under-reporting, and measurement flaws. These limitations could open interesting avenues for further research. Despite these limitations, this study provided insight into the use of framing message in the context of preschool children’s obesity health education among children’s caregivers in China.

## 5. Conclusions

In this cross-sectional study among 505 caregivers of preschool children in urban kindergartens, the gain-framed message and present-oriented message were most salient for stimulating the majority of caregivers to accept health messages. This research further supports the view that in the use of framing effects, the goal framing and the temporal framing can be used interactively. However, we should strictly control the sentence structure when compiling a health framing message. We speculate that the use of negative words in the message may affect the goal framing effects, and the quantity of messages may affect the temporal framing effects. However, the specific impact process should be explored in future studies. Other factors, such as gender, education background, participation in a health intervention, and career category, which might affect the framing effects, can help us release targeted health messages for specific groups of people to improve their acceptance of the message. 

## Figures and Tables

**Figure 1 ijerph-17-00770-f001:**
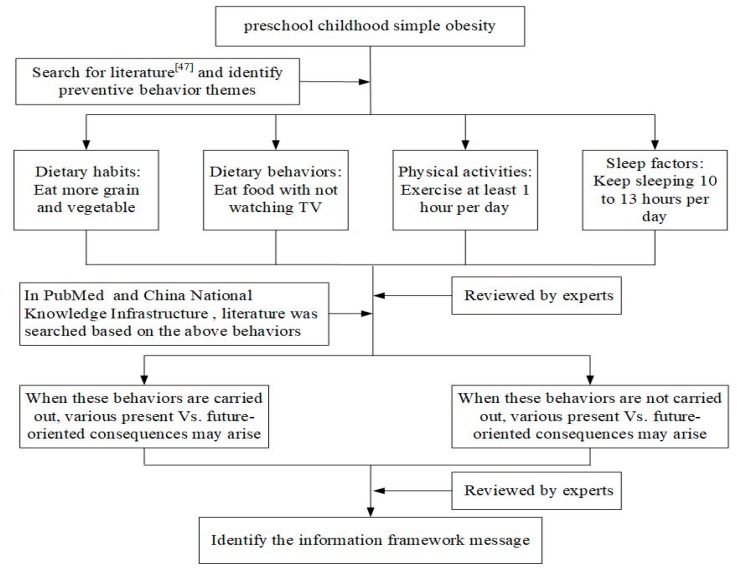
The framing messages’ design process.

**Table 1 ijerph-17-00770-t001:** Demographic characteristics of caregivers by source (*n* = 505).

Variables	Total (*n* = 505)	Online (*n* = 283)	Offline (*n* = 222)
	*n* (%)	*n* (%)	*n* (%)
Relationship			
Parents	409 (81.0%)	265 (93.6%)	144 (64.9%)
Grandparents	96 (19.0%)	18 (6.4%)	78 (35.1%)
Gender			
Female	412 (81.6%)	245 (86.6%)	167 (75.2%)
Male	93 (18.4%)	38 (13.4%)	55 (24.8%)
Nationality			
Han Nationality	485 (96.0%)	266 (94.0%)	219 (98.6%)
Minority Nationality	20 (4.0%)	17 (6.0%)	3 (1.4%)
Participated in health related intervention before ?			
Yes	125 (24.8%)	85 (30.0%)	40 (18.0%)
No	380 (75.2%)	198 (70.0%)	182 (82.0%)
Education Background			
Primary School and below	35 (6.9%)	1 (0.4%)	34 (15.3%)
Junior High School	70 (13.9%)	14 (4.9%)	56 (25.2%)
High School/Technical secondary School	103 (20.4%)	40 (14.1%)	63 (28.4%)
College/University Degree	258 (51.1%)	193 (68.2%)	65 (29.3%)
Postgraduate and above	39 (7.7%)	35 (12.4%)	4 (1.8%)
Career Category			
Administrative Organs, Soldiers, Teacher, Medical Staff, Scientist	166 (32.9%)	133 (47.0%)	33 (14.9%)
farmer	18 (3.6%)	5 (1.8%)	13 (5.8%)
Worker	91 (18.0%)	21 (7.4%)	70 (31.6%)
Commerce	124 (24.5%)	81 (28.6%)	43 (19.3%)
Retire	37 (7.3%)	15 (5.3%)	22 (9.9%)
Unemployed	69 (13.7%)	28 (9.9%)	41 (18.5%)
Monthly children’s dietary expenses (1 USD ≈ 7RMB)			
<¥500	103 (20.4%)	44 (15.5%)	59 (26.6%)
¥500–¥1000	260 (51.5%)	148 (52.3%)	112 (50.4%)
¥1001–¥1500	94 (18.6%)	60 (21.2%)	34 (15.3%)
>¥1500	48 (9.5%)	31 (11.0%)	17 (7.7%)

**Table 2 ijerph-17-00770-t002:** Caregivers’ choices of framing messages (*n* = 505).

Theme	Framing Type			
	GP Message	GF Message	LP Message	LF Message
Dietary Habits	385 (76.2%)	43 (8.5%)	63 (12.5%)	14 (2.8%)
Dietary Behaviors	182 (35.8%)	11 (2.4%)	253 (50.1%)	59 (11.7%)
Physical Activities	304 (60.2%)	125 (24.8%)	38 (7.5%)	38 (7.5%)
Sleep Factors	276 (54.7%)	177 (35.0%)	30 (5.9%)	22 (4.4%)

**Table 3 ijerph-17-00770-t003:** Chi-square test and pair-wise comparison of the results.

Goal Framing Effects					
Variables	Gain-Framed Message *n* (%)	Loss-Framed Message *n* (%)	Total *n* (%)	χ^2^	*p*
Dietary Habits (Dh)	428 a (84.8)	77 a (15.2)	505 (100.0)	467.203	0.000 **
Dietary Behaviors (Db)	193 b (38.2)	312 b (61.8)	505 (100.0)		
Physical Activities (Pa)	429 a (85.0)	76 a (15.0)	505 (100.0)		
Sleep Factors (Sf)	453 a (89.7)	52 a (10.3)	505 (100.0)		
**Temporal framing effects**					
**Variables**	**Present** **-oriented message**	**Future-oriented message**	**Total** ***n* (%)**	**χ^2^**	***p***
Dietary Habits (Dh)	448 a (88.7)	57 a (11.3)	505 (100.0)	155.575	0.000 **
Dietary Behaviors (Db)	434 a (85.9)	71 a (14.1)	505 (100.0)		
Physical Activities (Pa)	342 b (67.7)	163 b (32.3)	505 (100.0)		
Sleep Factors (Sf)	306 b (60.6)	199 b (39.4)	505 (100.0)		

Each letter (a, b) represents a subset of the framing messages selection. Statistical differences were observed in the selection ratio of the framing messages between two groups when the revised test level was α = 0.0083. ** *p* < 0.001 (statistically significant).

**Table 4 ijerph-17-00770-t004:** Binary logistic regression for goal framing.

Parameter		SE	Wald	OR	95% CI		*p*
Relationship	Parents	0.316	0.021	1.047	0.514	1.773	0.884
	Grandparents (ref.)						
Gender	Female	0.167	4.056	0.715	0.515	0.991	0.044 *
	Male (ref.)						
Nationality	Han Nationality	0.313	0.323	0.837	0.446	1.368	0.570
	Minority Nationality (ref.)						
Participated in related health intervention before	Yes	0.147	0.780	1.137	0.853	1.505	0.377
	No (ref.)						
Education Background		0.087	8.481	0.784	0.663	1.012	0.004 *
Career Category	Administrative organs, soldiers, teacher, medical staff, scientist	0.219	0.360	1.140	0.571	1.346	0.548
	farmer	0.370	0.728	0.728	0.351	1.473	0.394
	Worker	0.226	3.209	1.501	0.965	2.318	0.073
	Commerce	0.214	0.980	1.236	0.891	1.879	0.322
	Retire	0.360	1.090	1.456	0.721	2.937	0.297
	Unemployed (ref.)						
Monthly Children’s dietary expenses (1 USD ≈ 7RMB)		0.075	0.343	1.045	0.902	1.108	0.558
Age		0.011	0.791	1.010	0.969	1.022	0.374
Theme	Dietary Behaviors	0.162	166.285	0.124	0.096	0.173	0.000 **
	Physical Activities	0.187	1.185	1.225	0.843	1.735	0.276
	Sleep Factors	0.205	9.431	1.881	1.271	2.796	0.002 *
	Dietary Habits (ref.)						
Whether using negative words	Yes	0.138	71.492	0.321	0.238	0.409	0.000 **
	No (ref.)						
Temporal Framing	Present-oriented	0.149	26.626	2.155	1.610	2.885	0.000 **
	Future-oriented (ref.)						
Survey Method	Online	0.125	2.034	1.195	0.936	1.526	0.154
	Offline (ref.)						

Binary logistic regression analysis. * *p* < 0.05 and ** *p* < 0.001 (statistically significant).

**Table 5 ijerph-17-00770-t005:** Binary logistic regression for temporal framing.

Parameter		SE	Wald	OR	95%CI		*p*
Relationship	Parents	0.281	0.027	1.047	0.603	1.757	0.870
	Grandparents (ref.)						
Gender	Female	0.144	0.833	1.140	0.860	1.512	0.361
	Male (ref.)						
Nationality	Han Nationality	0.285	0.037	0.946	0.604	1.848	0.847
	Minority Nationality (ref.)						
Participated in health related intervention before	Yes	0.128	8.409	0.690	0.545	0.903	0.004 *
	No (ref.)						
Education Background		0.078	0.292	0.958	0.822	1.121	0.598
Career Category	Administrative organs, soldiers, teacher, medical staff, scientist	0.205	1.420	0.783	0.525	1.174	0.233
	farmer	0.441	4.697	2.604	1.096	6.162	0.030 *
	Worker	0.203	1.395	0.787	0.528	1.174	0.238
	Commerce	0.202	0.434	0.876	0.587	1.301	0.510
	Retire	0.305	0.333	1.192	0.649	2.172	0.564
	Unemployed (ref.)						
Monthly children’s dietary expenses (1USD≈7RMB)		0.067	1.082	0.932	0.814	1.061	0.298
Age		0.010	2.780	0.983	0.962	1.003	0.095
Theme	Dietary Behaviors	0.224	1.090	1.264	0.826	1.951	0.296
	Physical Activities	0.172	61.351	0.259	0.186	0.368	0.000 **
	Sleep Factors	0.171	99.623	0.182	0.131	0.255	0.000 **
	Dietary Habits (ref.)						
Whether using negative words	Yes	0.188	2.474	1.344	0.930	1.944	0.116
	No (ref.)						
Goal Framing	Gain-framed	0.187	26.058	2.602	1.802	3.756	0.000 **
	Loss-framed (ref.)						
Survey Method	Online	0.113	0.501	1.083	0.868	1.353	0.479
	Offline (ref.)						

Binary logistic regression analysis. * *p* < 0.05 and ** *p* < 0.001 (statistically significant).

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
