# Peer review of "Goal-Framing and Temporal-Framing: Effects on the Acceptance of Childhood Simple Obesity Prevention Messages among Preschool Children’s Caregivers in China"

_ijerph, 2020, doi:10.3390/ijerph17030770_

Round 1

Reviewer 1 Report

The authors investigate the interesting and important theme.

I have some questions about this article.

The introduction is too long. I recommend to some part of introduction to move the discussion section.

(eg. results of previous studies)

In Introduction

Line 43: missing reference
Line 44: missing reference

In Method

The item questionnaire was developed by investigators based on literature.
Did the authors check the validation of new-developed questionnaire?

Was the questionnaire  used in this study validated? ( for example..Chronbach Alpha Coefficient for test-retest)

In Result.

The participants were survey by both online and offline.
I recommend to compare the demographic characteristic between the two groups.

Were there differences between two groups?

This means the method of survey could affect the study's result.

Reviewer 2 Report

This is a good research which presents the results of investigation on caregivers’ acceptance of simple obesity prevention messages. The goal framing effects and temporal effects, together with their determining factors, were investigated. However, there are several weakness contained in this manuscript. I am really much concerned about the methodological aspect. Most importantly, the conceptual framework of this study lacks of theoretical support, and the discussion was not well presented. I therefore suggest making a major revision.

In the abstract, please concretely indicate how framing messages were developed? What concepts or theory were used to develop those messages? Please exclude “available literature”.

Page 2 line 49. What is BMI stand for? It has never mentioned before.

In the introductory part, please clearly indicate the age defined as preschool stage with citation(s).

Page 2 line 68, “Caregivers with different cognitive abilities and from different professional education levels may have considerable differences in the degree and the time they need for acceptance.” Please insert citation(s).

In the introduction, please add a paragraph presenting the objectives of this study, possibly the last paragraph. In addition, it would be more appropriate to present the situation relevant to simple obesity among preschool children in the case study as well.

Regarding the study design, section 2.1, it was mentioned that several factors were investigated on their influence on acceptance of framing messages. Please clearly indicate those factors and discuss on their possible effect on individual acceptance of framing messages, both goal framed and temporal framed. The discussions must be based on literature reviews. For instance, how gender, education level, participation in health training, etc. could influence individual acceptance of framing messages? Following these discussions, research hypotheses can be indicated. Without this part, the paper would definitely lack of scientific soundness, and would finally lead unconvincing findings.

 Page 2, line 119-120, it is not necessary to explain about WeChat.

In the section of participant selection, please explain how the size of population was decided.

In the section of questionnaire, it was indicated that the developed questionnaire was tested before the real survey. Please clearly report who and how many people participate in the test, and please report the results of pre-test. Normally, reliability of the developed questionnaire items must be reported.

Most importantly, I would suggest showing questionnaire items and a response category. Another option, please explain how participants were asked; so that, readers can realize characteristics of collected data. And please clearly explain how questionnaire items were developed. Please avoid too board explanations. For instance, “The item questionnaire was developed with reference to available literature”.

Regarding data collection, normally, it should be presented the date and place that data were collected.

Page 5, line 167, What do you mean by “initial survey”?

In table 1, Please eliminate the category “other relationship”.

Page 9, line 223, ‘’only 253 (50.1%) caregivers found the LP message acceptable.” 50.1% is the highest proportion. Please exclude “only”.

Page 9, line 226 regarding this statement “However, we found that the effect of framing effects affected by some factors through further analysis”, what do you mean?

Page 9, Line 257-259, it was indicated that “The results show that caregivers were more likely to accept the future-oriented message in Pa and Sf theme compared with Dh theme.” This is not true. According to Table 3, participants were more likely to accept present-oriented message in all themes.

Regarding the discussion part, it was well structured. Many findings were not discussed, contained no explanations. Please provide discussions on these findings, based on relevant studies and concepts.

The result in Table 2 showed that GP message was most preferred by participants for communicating about Dh, Pa, and Sf, except Db. Why? For the table 3, for communicating about Dh,Pa, and Sf, participants more accepted gain-framed message. Why? For communicating about Db, participants more accepted loss-framed message. Why? Many significant variables presented in Table 4 and 5 were not discussed on their influence on goal framing and temporal framing effects. For instance, influence of gender on participants’ acceptance of goal framed message. What presented in “Line 268-282” were quite board, and provide nothing for the contribution of this research. Please deliberatively and carefully discuss the results, and provide possible contributions based on those findings.

Minor issue

Page 2line48, Should be Geserick et al. [3]. Similarly, line 78-79. Myers [22]. Citations in many parts should be revised as well

Author Response

Please see the attachment.Author's response

Round 2

Reviewer 1 Report

Thank you for appropriate revision.

Reviewer 2 Report

The author(s) did not response to some important comments.

As I mentioned, regarding the study design, section 2.1, it was mentioned that several factors were investigated on their influence on acceptance of framing messages. Please clearly indicate those factors and discuss on their possible effect on individual acceptance of framing messages, both goal framed and temporal framed.

I also suggested author(s) doing more literature reviews in order to support the conceptual framework of this study.

Author(s) just put all variables in the analysis without any theoretical supports. I also don't think it makes sense to put "survey methods including online and offline in the analysis". I am not convinced with the study framework because of lack of theoretical discussions. 

Based on only this reason, this paper can be rejected.

In the section of data collection, It is also hard to follow. It is not so clear hon how each variable was measured.

Additionally, the statistic was not properly reported. I suggest following APA style.